# Signatures of HIV and Major Depressive Disorder in the Plasma Microbiome

**DOI:** 10.3390/microorganisms11041022

**Published:** 2023-04-14

**Authors:** Bryn C. Taylor, Mohammadsobhan Sheikh Andalibi, Stephen Wandro, Kelly C. Weldon, Gregory D. Sepich-Poore, Carolina S. Carpenter, Serena Fraraccio, Donald Franklin, Jennifer E. Iudicello, Scott Letendre, Sara Gianella, Igor Grant, Ronald J. Ellis, Robert K. Heaton, Rob Knight, Austin D. Swafford

**Affiliations:** 1Biomedical Sciences Graduate Program, University of California San Diego, San Diego, CA 92093, USA; 2Departments of Neurosciences and Psychiatry, HIV Neurobehavioral Research Center, University of California, San Diego, CA 92093, USA; mandalibi@health.ucsd.edu (M.S.A.);; 3Center for Microbiome Innovation, University of California San Diego, San Diego, CA 92093, USA; 4Skaggs School of Pharmacy and Pharmaceutical Sciences, University of California San Diego, San Diego, CA 92093, USA; 5Department of Bioengineering, University of California San Diego, San Diego, CA 92093, USA; 6Department of Psychiatry, School of Medicine, University of California, San Diego, CA 92093, USA; 7Departments of Medicine and Psychiatry, University of California San Diego, San Diego, CA 92093, USA; 8Division of Infectious Diseases and Global Public Health, University of California San Diego, San Diego, CA 92093, USA; 9Department of Pediatrics, School of Medicine, University of California San Diego, San Diego, CA 92093, USA; 10Department of Computer Science and Engineering, University of California San Diego, San Diego, CA 92093, USA

**Keywords:** depression, HIV, plasma microbiome, shotgun metagenomics

## Abstract

Inter-individual differences in the gut microbiome are linked to alterations in inflammation and blood–brain barrier permeability, which may increase the risk of depression in people with HIV (PWH). The microbiome profile of blood, which is considered by many to be typically sterile, remains largely unexplored. We aimed to characterize the blood plasma microbiome composition and assess its association with major depressive disorder (MDD) in PWH and people without HIV (PWoH). In this cross-sectional, observational cohort, we used shallow-shotgun metagenomic sequencing to characterize the plasma microbiome of 151 participants (84 PWH and 67 PWoH), all of whom underwent a comprehensive neuropsychiatric assessment. The microbial composition did not differ between PWH and PWoH or between participants with MDD and those without it. Using the songbird model, we computed the log ratio of the highest and lowest 30% of the ranked classes associated with HIV and MDD. We found that HIV infection and lifetime MDD were enriched in a set of differentially abundant inflammatory classes, such as Flavobacteria and Nitrospira. Our results suggest that the circulating plasma microbiome may increase the risk of MDD related to dysbiosis-induced inflammation in PWH. If confirmed, these findings may indicate new biological mechanisms that could be targeted to improve treatment of MDD in PWH.

## 1. Introduction

People infected with HIV (PWH) experience chronic immune activation and persistent inflammation, which are associated with increased risk of mortality and morbidities, even with virally suppressive antiretroviral therapy (ART) [1]. Multiple mechanisms have been proposed for these complications, including changes in the gut microbial community associated with HIV-induced autoimmune response and ART [2,3]. Recent work suggests that gut mucosal barrier dysfunction may lead to increased microbial translocation from the gastrointestinal (GI) tract into the blood, driving sustained inflammation [4]. The gut microbiome may also influence blood–brain barrier (BBB) integrity. Gut–brain axis pathways [5,6] have also been implicated in the pathogenesis of depression, which is common in PWH [7,8,9]. Together, gut microbiome changes and gut mucosal barrier dysfunction render PWH more exposed to pro-inflammatory microbial products, including lipopolysaccharide (LPS) and flagellin, which seem to play a role in neuroinflammation in the central nervous system (CNS) [10].

The linkage between depression and HIV has been well-documented [11,12]. Some studies suggest that the gut microbiota and its metabolites may contribute to depression and mood disorders via short-chain fatty acids (SCFAs), one of the gut microbiota’s metabolites [13]. However, due to insufficient data, this area is understudied, particularly with regard to microbial composition and its relationship with depression in PWH. We previously found that the fecal enrichment of *Enterobacteriaceae, Alistepes onderdonkii*, Bacteroides, and Parabacteroides distasonis was higher in HIV and HCV co-infected persons with lifetime major depressive disorder (MDD) compared to coinfected persons without MDD [9]. In another study, we observed differences in the relative abundance of colonic butyrate-producing bacterial species, such as Roseburia intestinalis in PWH [14].

Several studies investigated the association of gut microbial composition and depression [15,16]. However, studies connecting depression with the blood microbiome in PWH using new metagenomic approaches are lacking.

Previous studies examined the microbial contents in plasma using 16S qPCR and 16S rRNA gene amplicon sequencing. We recently developed a low-biomass-compatible, shallow-shotgun metagenomic sequencing protocol that enabled us to characterize associations between cancer and microbial DNA in circulation [17] and new compositionally coherent metagenomic analysis tools [18,19,20,21] to enable accurate taxonomic and functional microbial profiles. Here, we sought to apply these new methods to better characterize the interplay between microbial translocation into circulation, HIV infection, and MDD. We hypothesize that the circulatory microbiota dysbiosis in PWH will be linked to MDD.

## 2. Materials and Methods

### 2.1. Study Design and Participants

This cross-sectional observational cohort study was conducted among 151 participants (84 PWH and 67 people without HIV (PWoH)), who were recruited from the UC San Diego HIV Neurobehavioral Research Program (HNRP, https://hnrp.hivresearch.ucsd.edu/, 2 March 2023), and agreed to undergo standardized evaluations of MDD diagnosis using DSM-IV criteria. HIV infection was confirmed by MedMira point-of-care test (Halifax, Canada). Participants with incomplete clinical evaluations were excluded, as were individuals receiving antimicrobial therapy other than ART, which can alter the microbial profile. Participants who were current substance abusers and those with active psychiatric or neurological disorders that may interfere with participation in study protocols (e.g., poorly controlled epilepsy or psychosis) also were excluded.

After obtaining plasma, we performed 16S qPCR and shallow-shotgun sequencing to assess microbial abundance and community composition in circulating DNA from plasma samples and grouped them according to their combined HIV infection status and lifetime history of MDD. The UCSD Human Research Protections Program (irb.ucsd.edu) approved all study procedures, and all participants provided written informed consent.

### 2.2. Neuromedical and Laboratory Assessment

A detailed neuromedical assessment was used to collect a medical history that included use of ART and other medications, data to determine Centers for Disease Control (CDC) HIV disease staging, and biospecimen collection. Clinical chemistry panels, complete blood counts, rapid plasma reagin, and CD4+ T cells (flow cytometry) were performed at a Clinical Laboratory Improvement Amendments (CLIA)–certified laboratory. HIV RNA was measured in plasma using reverse transcription polymerase chain reaction (Amplicor, Roche Diagnostics, Indianapolis, IN, USA, with a lower limit of quantitation 40 copies/mL).

### 2.3. Evaluation of Depression

Using the computer-assisted Composite International Diagnostic Interview (CIDI, version 2.1. Geneva, Switzerland, World Health Organization), we evaluated Lifetime MDD according to DSM-IV. The Beck Depression Inventory-II was used to assess current depressed mood [22].

### 2.4. Protocol Details on the 16S qPCR

An initial volume of 250 µL of plasma was diluted by the addition of 750 µL of filter-sterilized phosphate-buffered saline, and the total cell-free DNA was extracted using the QIAamp Circulating Nucleic Acid Kit (QIAGEN, Valencia, CA, USA). Extracted DNA was further purified using AMPure XP SPRI paramagnetic beads (Beckman Coulter). For the quantitative detection of bacterial DNA via qPCR, we used the degenerated forward primer 341F (5′- CCTACGGGNGGCWGCAG-3′) and reverse primer 805R (5′-GACTACHVGGGTATCTAATCC-3′), targeting the regions V3–V4 of the 16S rRNA gene. For the quantitative analysis of human DNA, we used a primer set (forward: 5′-CAGTCTCACCTTCAACCG-3′; reverse: 5′-GTTATGTGCACACATGCTAC-3′) targeting the telomerase reverse transcriptase (tert) gene, reported to be a single-copy gene. Each amplification reaction was performed in a total volume of 10 µL with 5 µL of SsoAdvanced Universal SYBR Green Supermix (Bio-Rad Laboratories, Inc., Hercules, CA, USA), 0.5 µM of each primer (IDT), 0.25 µL of dsDNase and 0.25 µL of dithiothreitol (PCR Decontamination Kit (Enzo Life Sciences, Inc., Farmingdale, NY, USA), and 1 µL of purified genomic DNA. Standard curves for qPCR data analysis were generated based on serial ten-fold dilutions (from 5 ng/µL to 0.0005 ng/µL) of ZymoBIOMICS Microbial Community DNA Standard (Zymo Research, Irvine, CA, USA) and of a human genomic DNA (Promega Corporation, Madison, WI, USA).

### 2.5. Profiling of the Plasma Microbiome Composition Using Shallow-Shotgun Sequencing

Circulating DNA from frozen plasma was processed for shallow-shotgun sequencing [23] and mapped to the Web of Life database [24] using Bowtie 2 [25], followed by stringent decontamination in R using decontam (*p** = 0.5; prevalence and frequency modes) [26] as described [17]. The resulting BIOM tables were processed in QIIME 2 v2020.2 [27] for phylogenetic and non-phylogenetic alpha and beta diversity analyses and visualization. Between-group differences based on robust Aitchison distances [19] were tested using permutational multivariate analysis of variance (PERMANOVA) [28]. Alpha diversity between groups (Shannon diversity [29]) was compared with a Kruskal–Wallis test.

### 2.6. Differential Abundance Analysis

Songbird [18] and Qurro [20] were used to examine differences in specific groups of taxa collapsed to class in the plasma microbiome. Songbird is a compositionally aware differential abundance method that provides rankings of features (genomic operational taxonomic units [gOTUs]) based on their log fold-change with respect to covariates of interest. Here, the model formula described combined lifetime MDD and HIV status (“C(descriptive_group, Treatment(’HIV−/MDD−’)”, where descriptive_group is: HIV−/MDD−, HIV−/MDD+, HIV+/MDD−, HIV+/MDD+). Evaluation of the songbird model against baseline models obtained pseudo-Q2 values > 0, suggesting the model was not overfit. We then used Qurro to compute the log ratios of features that were identified by songbird to be associated with (1) HIV status (Figure 1a), (2) lifetime history of MDD (Figure 1b), as well as (3) LPS-producing microbes (Figure 1c). This approach was implemented to minimize the load of unknown taxa in each sample. In addition, using the log of the ratios of taxa provides equal weight to the relative changes of taxa [18]. The top and bottom 25% of feature ranks enabled us to retain 94% of the samples for comparisons. T tests and Cohen’s d were calculated to assess the significance (alpha = 0.05) and effect size of the log ratios of features.

## 3. Results

### 3.1. Demographic and Clinical Characteristics

HIV/MDD group comparisons on demographics and HIV disease and treatment characteristics are summarized in Table 1. PWoH with MDD are more likely to be male or white than the PWH and MDD group (X2 = 6.402, *p* = 0.01; X2 = 4.949, *p* = 0.03; respectively), and more likely to be heterosexual than PWH (X2 = 28.947, *p* < 0.0001). The duration of HIV infection is also longer in individuals with lifetime MDD than those without MDD (Student’s *t*-test, *p* = 0.04). PWH and lifetime MDD have significantly higher BDI-II scores than both PWH and PWoH without MDD (Table 1). We found no evidence for increased bacterial load between PWoH and PWH (Student’s *t*-test, *p* = 0.91), between people with or without MDD (Student’s *t*-test, *p* = 0.18), or combinations of these conditions (one-way ANOVA, *p* = 0.28), indicating no difference in microbial translocation based on MDD or HIV infection status for this cohort.

**Figure 1 microorganisms-11-01022-f001:**
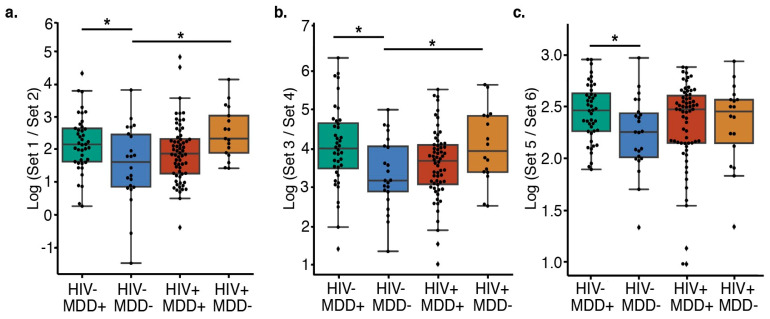
Comparison of the relative abundance of the taxa using log ratios of the ranked taxa sets in different subgroups. (**a**) Features distinguishing PWH from PWoH (*t*-test, HIV−/MDD− versus HIV−/MDD+ Cohen’s D = −0.63, BH-corrected *p* = 0.023; HIV−/MDD− versus HIV+/MDD− Cohen’s D = −0.95, BH-corrected *p* = 0.020) and (**b**) features distinguishing MDD+ from MDD− (*t*-test, HIV−/MDD− versus HIV−/MDD+ Cohen’s D = −0.70, BH-corrected *p* = 0.038; HIV−/MDD− versus HIV+/MDD− Cohen’s d = −0.75, BH-corrected *p* = 0.043). (**c**) HIV-/MDD- have a lower ratio of LPS producers than HIV−/MDD+ (Cohen’s D = −0.66, BH-corrected *p* = 0.028). (**a**–**c**) Center lines represent the mean, the top of the box represents the upper quartile, the top whisker represents the maximum value (excluding outliers), the bottom of the box represents the lower quartile, and the lower whisker represents the minimum value (excluding outliers). * Indicates significant differences (BH-corrected *p* < 0.05) between groups. The list of features in each set are present in Table 2.

**Table 2 microorganisms-11-01022-t002:** Sets of identified taxa used in log ratio computations using Songbird method.

Feature Group	Sets	Taxa
HIV features	Set 1 top 25%	Cytophagia, Flavobacteriia, Sphingobacteriia, Bacilli, Clostridia, Nitrospira, Acidithiobacillia
	Set 2 bottom 25%	Thermoprotei, Bacteroidia, Negativicutes, Fusobacteriia, Planctomycetia, Spirochaetia; phylum Cyanobacteria
MDD features	Set 3 top 25%	Blastocatellia, Flavobacteriia, Nitrospira, Alphaproteobacteria, Deltaproteobacteria, Epsilonproteobacteria; phylum Cyanobacteria
	Set 4 bottom 25%	Thermoprotei, Coriobacteriia, Bacteroidia, Cytophagia, Negativicutes, Spirochaetia, Opitutae

### 3.2. Composition of Microbial Communities within HIV and MDD Subgroups

Since MDD-associated inflammation may result from an altered microbial composition rather than quantity, we examined whether differences in the diversity of the plasma microbiome community as a whole were evident. As Figure 2 illustrates, the principal coordinate analysis (PCoA) plots reveal that the beta diversity is not different between PWH and PWoH based on robust principal component analysis (RPCA) [19] distances (PERMANOVA pseudo-F-statistic (pseudo-F) = 0.378, Benjamini–Hochberg corrected *p*-value (BH) = 0.923, Figure 2a). In addition, there is no significant difference in the α-diversity assessed by the Shannon index between the PWH and PWoH, suggesting that no community-wide difference is observed between the microbiome alpha diversity composition of the PWH and PWoH (Kruskal–Wallis (K–W) H = 0.045, BH = 0.972, Figure 2b). Beta diversity (RPCA PERMANOVA pseudo-F = 0.136, BH *p* = 0.923) and alpha diversity (Shannon index, K–W H = 0.522, BH *p* = 0.851) also do not differ between participants with or without MDD (Figure 2c,d), which, likewise, indicates no community-wide difference between the participants with and without MDD. When stratifying by both MDD and HIV status, neither beta diversity (RPCA PERMANOVA pseudo-F = 0.053, BH *p* = 0.652) nor alpha diversity (Shannon index, K–W H = 0.668, BH *p* = 0.972) differ among the four groups (Figure 2e,f).

The absence of conserved community-wide differences within the groups prompted us to examine whether specific taxa that would contribute to the pro-inflammatory state in PWH or MDD could still be enriched. We computed the log ratio of the highest 25% (“Set 1”, Table 2) and lowest 25% (“Set 2”, Table 2) of the ranked classes associated with HIV status using songbird [18] and visualized these ranks using Qurro [20] (Materials and Methods). In this case, the formula we used described whether the individual had HIV or not. There are significant differences in the log ratios of features that distinguish PWH from PWoH individuals amongst the MDD and HIV infection groups (Figure 1A). PWH or those who had MDD have higher log ratios of these features than the double-negative controls. Using pair-wise comparisons, we observe the significant differences in log ratios of features in PWoH without MDD (double-negative) group compared with PWH without MDD (Cohen’s D = −0.95, BH-corrected *p* = 0.020) and PWoH and MDD groups (Cohen’s D = −0.63, BH-corrected *p* = 0.023). These findings indicate that both PWH and individuals with MDD groups are enriched in taxa from Set 1 (Table 2, Figure 1a), which indicates that HIV and MDD are linked with an enriched set of differentially abundant microbes (gOTUs).

The log ratios differ of the highest 25% (“Set 3”, Table 2) and lowest 25% (“Set 4”, Table 2) of the ranked classes associated with MDD among the combined MDD and HIV infection groups (Figure 1b). In this case, the formula we used described whether the individual had MDD or not. We saw the same trend as before: that PWoH who are also without MDD (double-negative group) have a lower log ratio when compared to the other PWH (Cohen’s d = −0.75, BH-corrected *p* = 0.043) and MDD groups (Cohen’s D = −0.70, BH-corrected *p* = 0.038, Figure 1b), suggesting again that HIV and MDD are both associated with an enriched set of differentially abundant microbes (in this case, Set 3). Although the identification of particular plasma microbiome composition associated with these disease states is limited [30], we note that 71% of set 1 (5/7) and 100% of set 3 (7/7) are LPS-producing Gram-negative classes, including Flavobacteriia and Nitrospira, which are found in both sets. We, therefore, compared the log ratio of LPS-producing microbes (i.e., Gram-negative bacteria), which may contribute to inflammation [10], to non-LPS producing microbes (i.e., Gram-positive bacteria), but found that they are not significantly different across the HIV and MDD groups after multiple-testing correction (one-way ANOVA, *p* = 0.151). However, the LPS-producers are significantly increased in PWoH with MDD versus PWoH without MDD (double-negative group) by pairwise-comparison (Student’s *t*-test, t = 2.68, BH *p* = 0.028) (Figure 1c), indicating that PWoH with MDD are an enriched set of deferentially abundant gOTUs in LPS-producing ranked taxa. To examine whether persons with current versus lifetime MDD were driving this enrichment, we performed a subgroup analysis separating current and lifetime MDD individuals but were unable to obtain an accurate songbird model (pseudo-Q2 = −0.018), suggesting that no pattern of differential rankings is associated with this grouping.

## 4. Discussion

Previous studies linked gut microbiome changes to both HIV and depression [9,31]. However, to the best of our knowledge, our findings are the first study examining links between depression and the blood microbiome, and the first study examining the interaction of this microbiome community with both conditions. As methods for characterizing microbial signatures in blood in non-septic conditions have improved, the potential for extending microbial links beyond the gut compartment into the periphery have also expanded and provided a novel mechanism for examining the gut–brain axis.

Our results add to the growing evidence for the presence of a set of proinflammatory LPS-producing microbes whose nucleic acids are enriched in circulating plasma in HIV, with novel implications for individuals with MDD. The lack of significant difference in the ratio of LPS- and non-LPS-producing microbes between PWoH with MDD individuals and PWH individuals regardless of MDD status suggests that the blood microbiome profile of PWH and without MDD may have already shifted into a pathologic, depression-promoting state. In addition, the overlap of HIV-associated and MDD-associated microbial classes suggests that there may be shared underlying physiological conditions that contribute or are shaped by the development of this microbial profile. We observed no evidence for increased bacterial load and microbiome diversity between PWoH and PWH, between subjects with or without MDD, or combinations of these conditions, suggesting there is no difference in microbial translocation based on MDD or HIV infection status for this cohort, in contrast to previous studies that reported an increase. However, these prior reports primarily relied on measurements of plasma LPS, which would capture both translocated microbes as well as pro-inflammatory molecules entering circulation through the compromised mucosal barrier.

Most human microbiome studies used stool, saliva, or cervical–vaginal lavage fluid samples and there are limited studies on plasma microbiome, due to challenging technical demands. In this study, we implemented the shallow-shotgun metagenomic sequencing protocol. Of note, although some species can be rendered with 16S sequencing, 16S sequencing variable-region amplicon sequences often do not differentiate taxa below the genus or family level [32]. It is stated that shallow-shotgun sequencing is superior to 16S sequencing for recovery and annotation of species. In addition, shallow-shotgun sequencing renders more accurate species-level taxonomic resolution and functional profiles of the human microbiome studies than 16S sequencing [21].

We acknowledge some limitations in our study. The fecal microbiome could be assessed and compared to the plasma microbiome. Regarding our findings, we would evaluate both plasma and fecal microbiome composition in our future studies. In addition, it is argued that in studies on blood and biopsy specimens where there is low bacterial biomass, and the risk of host DNA contamination is relatively high, shallow-shotgun sequencing may not be a viable replacement for 16S sequencing [21]. The BDI-II scores are higher in both PWH and PWoH with lifetime MDD than those without, suggesting that those with lifetime MDD continue to suffer from a substantially increased burden of depressed mood at the time of their participation in this study.

It has been reported that chronic inflammation is associated with treatment-resistant depression. In addition, inflammation has been linked with more severe depression symptoms and more resistance to treatment in patients with MDD [33]. Strong evidence suggests that inflammation is involved in the pathogenesis of depression in PWH [34]. Given our identification of DNA from inflammation-inducing microbes in circulation and the efficacy of the antibiotic minocycline for this depression subtype [35], additional investigations into modulation of the microbial community to reduce inflammation are warranted, though this has not been studied in the context of HIV. We anticipate this will be an active area of future inquiries in the field. Furthermore, although HIV is readily diagnosable from a peripheral blood draw, depression is not, and the stigma surrounding its diagnosis, particularly in the already-stigmatized HIV population, leads to widespread underdiagnosis. The identification of specific ratios of microbes associated with depression suggests that a microbial-DNA-based marker for this condition, and other neurological conditions associated with BBB breakdown, warrant further investigation.

## Figures and Tables

**Figure 2 microorganisms-11-01022-f002:**
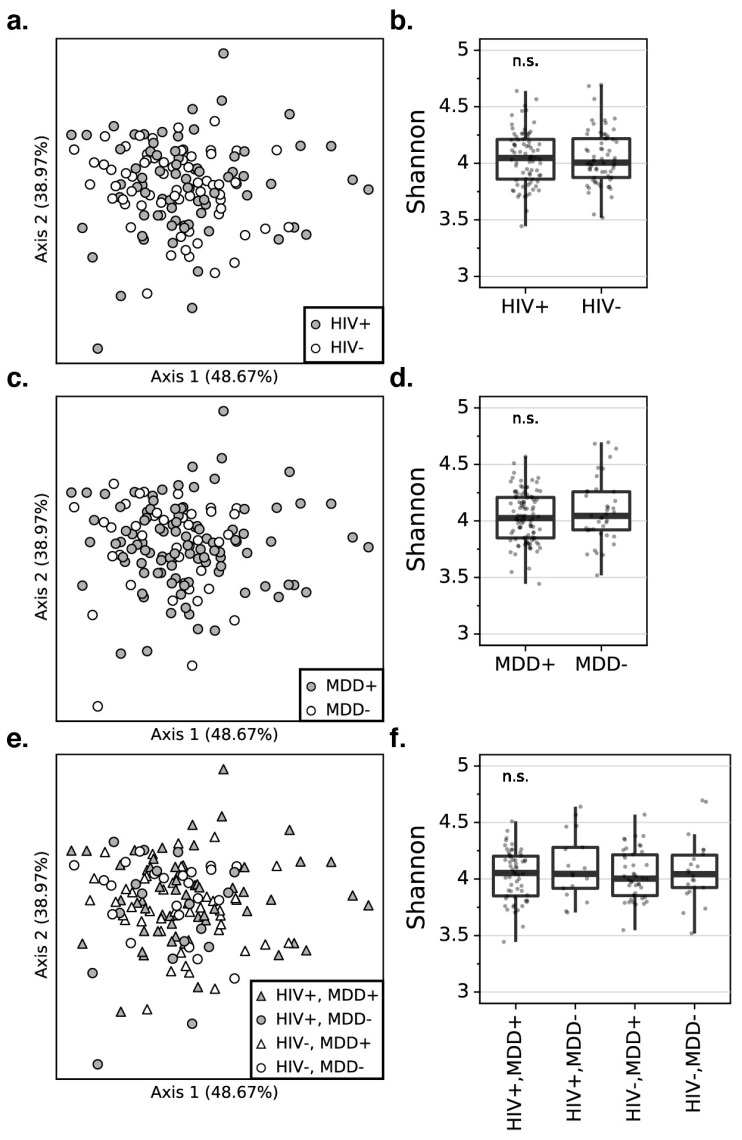
The PCoA plots for the robust principal component analysis (RPCA) distance matrix reveal that there is no significant difference in the beta diversity between PWH and PWoH (**a**), participants with and without MDD (**c**), and combined HIV and MDD groups (**e**). The percentage of the total variance is presented by each axis. The box plots represent that there is no significant difference in the alpha diversity using the Shannon index between PWH and PWoH (**b**), participants with and without MDD (**d**), combined HIV and MDD groups (**f**). PWH: people with HIV, PWoH: people without HIV, MDD: major depressive disorder.

**Table 1 microorganisms-11-01022-t001:** Comparison of demographic and clinical characteristics between HIV/MDD subgroups.

	HIV−/MDD− (a)	HIV−/MDD+ (b)	HIV+/MDD− (c)	HIV+/MDD+ (d)	Group Significance (α = 0.05)
N	23	44	18	66	
Age ^1^	45.7 (13.5)	45.0 (12.6)	44.5 (14.3)	46.1 (11.3)	
Education ^1^	14.4 (2.7)	13.6 (2.7)	13.8 (2.0)	13.4 (2.5)	
Male, n (%)	20 (87%)	30 (68%)	16 (89%)	58 (88%)	b < d
Ethnicity, n (%)					
African American	4 (17%)	4 (9%)	1 (6%)	13 (20%)	
Caucasian	14 (61%)	32 (73%)	10 (55%)	34 (51%)	b > d
Hispanic	4 (17%)	5 (11%)	5 (28%)	16 (24%)	
Other	1 (5%)	3 (7%)	2 (11%)	3 (5%)	
Estimate premorbid verbal IQ ^1^	103.2 (13.7)	103.9 (14.2)	100 (15.2)	100.4 (11.9)	
Sexual orientation, n (%)					
Bisexual	0 (0%)	8 (18%)	3 (18%)	6 (9%)	
Heterosexual	12 (55%)	28 (64%)	2 (12%)	13 (20%)	a, b > c, d
Homosexual	11 (45%)	8 (18%)	13 (70%)	47 (71%)	
AIDS, n (%)			6 (33%)	30 (45%)	
Estimated duration of infection (years) ^1^			8.5 (7.6)	13.1 (8.2)	c < d
Nadir CD4+ T-cell count ^2^			319 (105–5141)	234 (147–350)	
CD4+ T-cell count ^2^			660 (426–9361)	639 (492–7681)	
Undetectable plasma on ART, n (%)			16 (89%)	63 (95%)	
Undetectable CSF on ART, n (%)			18 (100%)	57 (87%)	
ART status, n (%)					
On ART			17 (94%)	64 (97%)	
ARV regimen type					
NNRTIs + NRTIs			7 (39%)	22 (33%)	
NRTIs + IIs			6 (33%)	12 (18%)	
NRTIs + PIs			1 (5%)	25 (38%)	
NRTIs			1 (5%)	2 (3%)	
Other			2 (11%)	3 (5%)	
Off ART			1 (6%)	2 (3%)	
Employed, n (%)	11 (45%)	19 (43%)	9 (50%)	26 (40%)	
Lifetime any substance diagnosis	12 (52%)	38 (86%)	9 (50%)	55 (66)	
Current substance use diagnosis	0 (0%)	0 (0%)	0 (0%)	0 (0%)	
Beck Depression Inventory-II ^1^	6.61 (8.83)	14.57 (12.59)	5.57 (5.91)	15.88 (10.57)	b > a, c; d > a, c
Microbial DNA concentration (ng/μL) ^2^	0.58 (0.21–0.99)	0.44 (0.21–0.95)	0.27 (0.22–0.43)	0.45 (0.31–0.82)	
% Microbial DNA ^2^	3.4 (2.3–4.51)	2.8 (2.0–4.8)	3.5 (2.2–6.5)	3.2 (2.3–4.5)	

Student’s *t*-tests were used for all normally distributed continuous variables (age, education, estimate verbal IQ, estimated duration of HIV infection; Wilcoxon tests were used for nadir CD4, current CD4, microbial DNA concentration, and % microbial DNA; chi-square tests used for all nominal variables (% Caucasian, % AIDS, % undetectable plasma on ART, % undetectable CSF on ART, % bisexual and/or homosexual, ART status, lifetime (LT) any substance diagnosis). ART: antiretroviral therapy, NNRTIs: non-nucleoside reverse transcriptase inhibitors, NRTIs: nucleoside/nucleotide reverse transcriptase inhibitors, IIs: integrase inhibitors, PIs: protease inhibitors. The final column, *Group Significance*, indicates if the groups show a significant difference (alpha = 0.05) and in which direction. ^1^ mean (SD); ^2^ median (IQR).

## Data Availability

The data generated in this study are available publicly in Qiita under the study ID 12667. Shotgun sequencing data associated with this study can be found under EBI accession number ERP119598.

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
