# Peer review of "Signatures of HIV and Major Depressive Disorder in the Plasma Microbiome"

_microorganisms, 2023, doi:10.3390/microorganisms11041022_

Round 1
Reviewer 1 Report
When I read the abstract I found the topic interesting and relevant. Well written design and methodology.
But the results section is hard to understand.
In my opinion, the work would be more informative if it communicated the results in a less technical and more interpreted way for better understanding.
For this purpose, I would recommend:
In the results section, I suggest writing the results of figure 1, clearly explaining their meaning for better understanding. For example, as a physician, I am interested in these results, but Figure 1 is explained as " Log ratios of aggregated bacterial relative abundances collapsed to class-level assignments ". It is confusing to understand the results they want to tell us. My recommendation is to write the Figure 1 results in the results section in a clear and less technical way for a better understanding.
By contrast, Figure 2 is a very confusing graph. My recommendation would be to clarify the explanatory titles of the chart (are axes 1 and 2 the best way to explain the chart?) and the chart legend. On the other hand, in relation to figure 2, I appreciate the translation of these data in the results section.
Regarding other recommendations, I would suggest describing the antiretroviral treatment that people with HIV received at the baseline characteristics because some of them, such as the NNITRs or the integrase inhibitors, have effects on the CNS.
The last part about the analyze of log rank using Songbird is also very confusing in interpreting the results they got.
In short, I think the idea of the study is novel and relevant, but the way the authors present the results is confusing and their interpretation difficult to understand. Perhaps the article is specialized and complex, but if the purpose of the publication is transmitting their results, I think the manuscript would be better if the authors interpreted the result more clearly.
Author Response
Manuscript ID microorganisms-2292885
Reviewer(s)' Comments:
Reviewer 1:
When I read the abstract I found the topic interesting and relevant. Well written design and methodology. But the results section is hard to understand.
- In my opinion, the work would be more informative if it communicated the results in a less technical and more interpreted way for better understanding.
For this purpose, I would recommend:
In the results section, I suggest writing the results of figure 1, clearly explaining their meaning for better understanding. For example, as a physician, I am interested in these results, but Figure 1 is explained as " Log ratios of aggregated bacterial relative abundances collapsed to class-level assignments ". It is confusing to understand the results they want to tell us. My recommendation is to write the Figure 1 results in the results section in a clear and less technical way for a better understanding.
Response:
We thank the reviewer for this valuable comment. We changed the Fig.1 legend and added some explanations to the result section. Please see Page 5, Line 190, and Page 6, Line 232 copied below:
Figure 1. Comparison of the relative abundance of the taxa using log ratios of the ranked taxa sets in different subgroups. A) Features distinguishing PWH from PWoH (t-test, HIV-/MDD- versus HIV-/MDD+ Cohen’s D = -0.63, BH-corrected p = 0.023; HIV-/MDD- versus HIV+/MDD- Cohen’s D = -0.95, BH-corrected p = 0.020) and B) Features distinguishing MDD+ from MDD- (t-test, HIV-/MDD- versus HIV-/MDD+ Cohen’s D = -0.70, BH-corrected p = 0.038; HIV-/MDD- versus HIV+/MDD-Cohen’s d = -0.75, BH-corrected p = 0.043). C) HIV-/MDD- had a lower ratio of LPS producers than HIV-/MDD+ (Cohen’s D = -0.66, BH-corrected p = 0.028). A-C) Center lines represent the mean, the top of the box represents the upper quartile, the top whisker represents the maximum value (excluding outliers), the bottom of the box represents the lower quartile, and the lower whisker represents the minimum value (excluding outliers). *Indicates significant differences (BH-corrected p < 0.05) between groups. The list of features in each set are present in Table 2.
Results:
The absence of conserved community-wide differences within the groups prompted us to examine whether specific taxa could still be enriched that would contribute to the pro-inflammatory state in PWH or MDD. We computed the log ratio of the highest 25% (“Set 1”, Table 2) and lowest 25% (“Set 2”, Table 2) of the ranked classes associated with HIV status using Songbird [18] and visualized these ranks using Qurro [20] (Materials and Methods). In this case, the formula we used described whether the individual had HIV or not. There were significant differences in the log ratios of features that distinguish PWH from PWoH individuals amongst the MDD and HIV infection groups (Fig. 1A). PWH or those who had MDD had higher log ratios of these features than did the double-negative controls. Using pair-wise comparisons, we observed the significant differences in log ratios of features in PWoH without MDD (double-negative) group compared with PWH without MDD (Cohen’s D = -0.95, BH-corrected p = 0.020) and PWoH and MDD groups (Cohen’s D = -0.63, BH-corrected p = 0.023). These findings are indicating that both PWH and individuals with MDD groups were enriched in taxa from Set 1 (Table 2, Fig. 1a), which indicating that HIV and MDD are linked with an enriched set of differentially abundant microbes (gOTUs).
The log ratios also differed of the highest 25% (“Set 3”, Table 2) and lowest 25% (“Set 4”, Table 2) of the ranked classes associated with MDD among the combined MDD and HIV infection groups (Fig. 1b). In this case, the formula we used described whether the individual had MDD or not. We saw the same trend as before: that PWoH who also were without MDD (double-negative group) had a lower log ratio when compared to the other PWH (Cohen’s d = -0.75, BH-corrected p = 0.043) and MDD groups (Cohen’s D = -0.70, BH-corrected p = 0.038, Fig. 1b), suggesting again that HIV and MDD are both associated with an enriched set of differentially abundant microbes (in this case, Set 3). Although the identification of particular plasma microbiome composition associated with these disease states is limited [30], we note that 71% of Set 1 (5/7) and 100% of Set 3 (7/7) are LPS-producing Gram-negative classes, including Flavobacteriia and Nitrospira which were found in both sets. We therefore compared the log ratio of LPS-producing microbes (i.e. Gram negative bacteria) which may contribute to inflammation [10], to non-LPS producing microbes (i.e. Gram positive bacteria) but found that they were not significantly different across the HIV and MDD groups after multiple-testing correction (one-way ANOVA, p = 0.151). However, the LPS-producers were significantly increased in PWoH with MDD versus PWoH without MDD (double-negative group) by pairwise-comparison (Student’s t-test, t = 2.68, BH p = 0.028) (Fig. 1c) indicating that PWoH with MDD were enriched set of deferentially abundant gOTUs in LPS-producing ranked taxa. To examine whether persons with current versus lifetime MDD were driving this enrichment, we performed a subgroup analysis separating current and lifetime MDD individuals but were unable to obtain an accurate Songbird model (pseudo-Q2 =-0.018), suggesting that no pattern of differential rankings was associated with this grouping.
==================================================================
- By contrast, Figure 2 is a very confusing graph. My recommendation would be to clarify the explanatory titles of the chart (are axes 1 and 2 the best way to explain the chart?) and the chart legend. On the other hand, in relation to figure 2, I appreciate the translation of these data in the results section.
Response:
We thank the reviewer for this comment. In general, in principal coordinate analysis (PCoA) plots, the scales of the horizontal and vertical axes are relative distances and have no practical meaning. In brief, PCoA1 (Axis 1) and PCoA2 (Axis 2) explained the proportion of the variance of the abundance of OTUs. The spatial distance of sample points represents the distance between samples.
We changed the Fig.2 legend and elaborated it on the result section. Please see Page 6, Line 188, and Page 7, Line 201 copied below:
The PCoA plots for the robust principal-component analysis (RPCA) distance matrix reveal that there is no significant difference in the beta diversity between PWH and PWoH (a), participants with and without MDD (c), and combined HIV and MDD groups (e). The percentage of the total variance is presented by each axis. The box plots represent that there is no significant difference in the alpha diversity using the Shannon index between PWH and PWoH (b), participants with and without MDD (d), combined HIV and MDD groups (f). PWH: People with HIV, PWoH: People without HIV, MDD: major depressive disorder.
Results:
As Fig. 2 illustrates, the principal coordinate analysis (PCoA) plots revealed that the beta diversity was not different between PWH and PWoH based on robust principal-component analysis (RPCA) distances (PERMANOVA pseudo-F-statistic (pseudo-F) = 0.378, Benjamini-Hochberg corrected p-value (BH)= 0.923, Fig. 1a). In addition, there was no significant difference in the α-diversity assessed by the Shannon index between the PWH and PWoH, suggesting that no community-wide difference was observed between the microbiome alpha diversity composition of the PWH and PWoH (Kruskal-Wallis (K-W) H = 0.045, BH = 0.972, Fig. 1b). Beta diversity (RPCA PERMANOVA pseudo-F = 0.136, BH p = 0.923) and alpha diversity (Shannon index, K-W H = 0.522, BH p = 0.851) also did not differ between participants with or without MDD (Fig. 2 c and d), which likewise indicates no community-wide difference between the participants with and without MDD.
- Regarding other recommendations, I would suggest describing the antiretroviral treatment that people with HIV received at the baseline characteristics because some of them, such as the NNITRs or the integrase inhibitors, have effects on the CNS.
Response:
We thank the reviewer for this important comment. We added the information regarding the participants’ ARV regimens to the Table 1. Please see Pages 4 and 5. The table is also copied below:
|
HIV-/MDD- (a) |
HIV-/MDD+ (b) |
HIV+/MDD- (c) |
HIV+/MDD+ (d) |
Group Significance (a= 0.05) |
|
|
n |
23 |
44 |
18 |
66 |
|
|
Age1 |
45.7 (13.5) |
45.0 (12.6) |
44.5 (14.3) |
46.1 (11.3) |
|
|
Education1 |
14.4 (2.7) |
13.6 (2.7) |
13.8 (2.0) |
13.4 (2.5) |
|
|
Male, n (%) |
20 (87%) |
30 (68%) |
16 (89%) |
58 (88%) |
b < d |
|
Ethnicity, n (%) |
|||||
|
African American |
4 (17%) |
4 (9%) |
1 (6%) |
13 (20%) |
|
|
Caucasian |
14 (61%) |
32 (73%) |
10 (55%) |
34 (51%) |
b > d |
|
Hispanic |
4 (17%) |
5 (11%) |
5 (28%) |
16 (24%) |
|
|
Other |
1 (5%) |
3 (7%) |
2 (11%) |
3 (5%) |
|
|
Estimate Premorbid Verbal IQ1 |
103.2 (13.7) |
103.9 (14.2) |
100 (15.2) |
100.4 (11.9) |
|
|
Sexual Orientation, n (%) |
|||||
|
Bisexual |
0 (0%) |
8 (18%) |
3 (18%) |
6 (9%) |
|
|
Heterosexual |
12 (55%) |
28 (64%) |
2 (12%) |
13 (20%) |
a, b > c, d |
|
Homosexual |
11 (45%) |
8 (18%) |
13 (70%) |
47 (71%) |
|
|
AIDS, n (%) |
6 (33%) |
30 (45%) |
|||
|
Estimated Duration of Infection (years)1 |
8.5 (7.6) |
13.1 (8.2) |
c < d |
||
|
Nadir CD4+ T-cell Count2 |
319 [105-5141] |
234 [147-350] |
|||
|
CD4+ T-cell Count2 |
660 [426-9361] |
639 [492-7681] |
|||
|
Undetectable Plasma on ART, n (%) |
16 (89%) |
63 (95%) |
|||
|
Undetectable CSF on ART, n (%) |
18 (100%) |
57 (87%) |
|||
|
ART Status, n (%) |
|||||
|
On ART |
17 (94%) |
64 (97%) |
|||
|
ARV regimen type |
|
|
|
|
|
|
NNRTIs + NRTIs |
|
|
7 (39%) |
22 (33%) |
|
|
NRTIs + IIs |
|
|
6 (33%) |
12 (18%) |
|
|
NRTIs + PIs |
|
|
1 (5) |
25 (38%) |
|
|
NRTIs |
|
|
1 (5%) |
2 (3%) |
|
|
Other |
|
|
2 (11%) |
3 (5%) |
|
|
Off ART |
|
|
1 (6%) |
2 (3%) |
|
|
Employed, n (%) |
11 (45%) |
19 (43%) |
9 (50%) |
26 (40%) |
|
|
Lifetime Any Substance Diagnosis |
12 (52%) |
38 (86%) |
9 (50%) |
55 (66) |
|
|
Current Substance Use Diagnosis |
0 (0%) |
0 (0%) |
0 (0%) |
0 (0%) |
|
|
Beck Depression Inventory-II1 |
6.61 (8.83) |
14.57 (12.59) |
5.57 (5.91) |
15.88 (10.57) |
b > a, c; d > a, c |
|
Microbial DNA Concentration (ng/μl)2 |
0.58 [0.21-0.99] |
0.44 [0.21-0.95] |
0.27 [0.22-0. 43] |
0.45 [0.31-0.82] |
|
|
% Microbial DNA2 |
3.4 [2.3-4.51 |
2.8 [2.0-4.8 |
3.5 [2.2-6.5] |
3.2 [2.3-4.5] |
===========================================================================
- The last part about the analyze of log rank using Songbird is also very confusing in interpreting the results they got.
In short, I think the idea of the study is novel and relevant, but the way the authors present the results is confusing and their interpretation difficult to understand. Perhaps the article is specialized and complex, but if the purpose of the publication is transmitting their results, I think the manuscript would be better if the authors interpreted the result more clearly.
Response:
We thank the reviewer for a thorough review of our study and his/her interest in our study. We did our best to interpret our results in detail and make the result section more clear.

Reviewer 2 Report
The paper by Andalibi et al. concerns a young and interesting research area that addresses the blood microbiome (or bacterial DNA in blood). First, this kind of analyses are technically very difficult to perform and this manuscript adds to technical progression in the field. Second, the blood microbiome might be in the causal pathway of diseases in relation to the gut microbiome and, therefore, this manuscript is of interest to the microbiome field. Third, the authors bring their conclusions with a certain amount of care (they are not too conclusive) which is in balance with the experimentation. Finally, the paper is well written and straightforward and I have only one minor comment: in line 63 of the introduction the authors state that associations between gut microbiome and depression are not robust. I do not agree as there are many studies now ranging from animal studies to meta-analyzed cohort studies that clearly show very robust and replicable associations.
Author Response
Manuscript ID microorganisms-2292885
Reviewer(s)' Comments:
Reviewer 2:
- The paper by Andalibi et al. concerns a young and interesting research area that addresses the blood microbiome (or bacterial DNA in blood). First, this kind of analyses are technically very difficult to perform and this manuscript adds to technical progression in the field. Second, the blood microbiome might be in the causal pathway of diseases in relation to the gut microbiome and, therefore, this manuscript is of interest to the microbiome field. Third, the authors bring their conclusions with a certain amount of care (they are not too conclusive) which is in balance with the experimentation. Finally, the paper is well written and straightforward, and I have only one minor comment: in line 63 of the introduction the authors state that associations between gut microbiome and depression are not robust. I do not agree as there are many studies now ranging from animal studies to meta-analyzed cohort studies that clearly show very robust and replicable associations.
Response:
We thank the reviewer for a thorough review of our study and his/her interest in our study. We edited the mentioned sentence Please see Page 2, Line 72, copied below:
Several studies investigated the association of gut microbial composition and depression [15, 16]. However, studies connecting depression with the blood microbiome in PWH using new metagenomic approaches are lacking.
